# Design of a Four-Axis Robot Arm System Based on Machine Vision

**Yijie Wang** [1,†]**, Yushan Zhou** [2,†]**, Lai Wei** [3] **and Ruiya Li** [4,*]

1    School of International Education, Wuhan University of Technology, Wuhan 430070, China; 324245@whut.edu.cn
2    School of Automation, Wuhan University of Technology, Wuhan 430070, China; 310601@whut.edu.cn
3    School of Information, Wuhan University of Technology, Wuhan 430070, China; laiwei@whut.edu.cn
4    School of Mechanical and Electronic Engineering, Wuhan University of Technology, Wuhan 430070, China
*    Correspondence: liruiya@whut.edu.cn
†    These authors contributed equally to this work.

**Abstract:** With the concept of industrial automation gradually being put forward, the four-axis robotic arm is gradually being applied in industrial production environments due to its advantages such as a stable structure, easy maintenance, and expandability. However, it is difficult to diversify and improve the traditional four-axis robotic arm system due to the high software and hardware coupling and the single system design, which results in high production costs. At the same time, its low intelligence and high-power consumption limit its wide application. The paper proposes an embedded design of a four-axis manipulator system based on vision guidance. Based on the robot kinematics theory and geometric principles, the dynamics simulation of the manipulator model is carried out. Through the forward and reverse analysis of the manipulator model and the trajectory planning of the manipulator, the YOLOV7 target detection algorithm is introduced and deployed on the embedded device, which greatly reduces the manufacturing cost of the manipulator while meeting the control and power consumption requirements. It has been verified by experiments that the robot arm in this paper can achieve an end accuracy of 0.05 mm under the condition of a load of 1 kg using the ISO 9283 international standard, and the recognition algorithm adopted can achieve a recognition accuracy of 95.2% at a frame rate of 29. The overall power consumption is also lower than that of traditional robotic arms.

**Keywords:** robotic arm system; path planning; vision guidance; depth detection; motion control system

## 1. Introduction

With the development of industrial automation, robotic arms have been widely used in production life instead of humans to perform high-risk and high-repetition work [1–3]. Compared with traditional robotic arms, four-axis robotic arms are gradually being used in industrial production scenarios due to their structural stability, convenience of maintenance, and scalability. In the past few years, research has mainly focused on the path planning algorithms for improving the intelligence of robotic arms [4,5], on the innovation of the robotic arm software and hardware system to reduce the cost and power consumption, and on the complete exploitation of the robotic arm system for improving the application scenario of the system, which has resulted in the development of robotic arm systems being improved [6]. However, as robotic arms are gradually being used in various industries, industrial automation has put forward higher requirements in terms of robotic arm intelligence and transplantation flexibility [7,8].

Cabre et al. proposed a case study of project-based learning developed at the University of Leyda in Spain [9]. This article combines the knowledge of computer vision and robot control to complete automation project tasks. The article develops a computer vision

system that must detect small objects randomly placed on a target surface and control an educational robotic arm to pick them up and move them to a predefined destination. The robotic arm system comprehensively uses computer vision and robotic arm control theory, but its automation level is low, and it cannot complete precise grasping autonomously. Sepulveda et al. proposed a dual-arm robot system [10] which combines the image segmentation algorithm with the dynamic programming algorithm and the occlusion algorithm to improve the picking success rate of the harvester. Image segmentation algorithms (based on an SVM pixel classifier, watershed transformation, and point cloud registration) are responsible for the detection and localization of eggplants. Experimental results show that the robotic arm can harvest 91.67% of the total number of eggplants under the proposed common scenario. Since the system uses computer image processing algorithms instead of deep learning algorithms to detect targets, the accuracy rate is low. Yang et al. introduced the development of a shared control system for intelligent manipulators [11]. The target object is detected by the vision system and then displayed to the user in a video, and through the analysis of the invoked EEG signal, a brain–computer interface is developed to infer the exact object the user needs. These results are then transmitted to a shared control system that enables precise object manipulation through visual servoing technology. Through the coordination of task motion and ego-motion (CTS) method, the robot has an autonomous obstacle avoidance function, which improves the intelligence of the shared control system. In this system, the basic color separation algorithm is used for the target detection algorithm, and it cannot recognize complex color objects independently.

For mobile robots to complete complex tasks such as explosive disposal using two dexterous hands [12], Sun et al. developed the impedance control approach with slippage detection by considering slippage tendency and slippage intensity to produce a stable in-hand manipulation. Furthermore, the Faster R-CNN is employed to determine the grasping region for robot manipulation through object detection and learning. Finally, the explosive disposal scene is designed to justify the effectiveness and good performance of the proposed methods. This robotic arm has a good motion effect, but it is difficult to promote it on a large scale because it is not very portable.

Du et al. proposed an offline–merge–online robot teaching method (OMORTM). Specifically [13], a virtual real fusion interactive interface (VRFII) was first developed by projecting a virtual robot into the real scene with an augmented-reality (AR) device, aiming to implement offline teaching. Second, a visual-aid algorithm (VAA) was proposed to improve offline teaching accuracy. Third, a gesture and speech teaching fusion algorithm (GSTA) with fingertip tactile force feedback was developed to obtain the natural teaching pattern and improve the interactive accuracy of teaching the real or virtual robot. This kind of robot combines vision algorithm and basic control of the robot, but research has not been conducted on the upper computer system, so it is difficult to popularize.

Luo et al. proposed a service-oriented multiagent system (SoMAS) for the control and analysis of the cyber-physical system (CPS) in manufacturing automation utilizing an on-contact dynamic obstacle avoidance, seven-DoF robot arm [14]. The interfaces of the services which the robot arm subsystem should provide to fully exploit its capability are identified. Specifically, the services of moving, object recognition, object fetching, and safety of human–robot interaction are considered the fundamental functionalities that the robot arm should provide. The way to evaluate the quality of services (QoS) for the robot arm subsystem is also explained. To build such a robot arm subsystem, the system architecture is proposed. Also, implementation for the subsystem, which includes 3D model-based object recognition, grasp database for object fetching, and online noncontact obstacle avoidance for the safety of human–robot interaction, is provided. The experimental results demonstrate that the capabilities of 3D model-based object recognition, object fetching, and dynamic collision avoidance are successfully implemented. This robot can realize object recognition and human–computer interaction, but it is not developed with an embedded system, so its power consumption is high and it is difficult for secondary development

To solve the above problems, a vision-guided intelligent robotic arm control system is proposed in this paper, including the overall mechanical structure of the arm and control system design, related algorithms, and upper computer software implementation, as shown in Figure 1. Firstly, simulations are conducted according to the kinematic theory of the robotic arm, the modeling and forward and reverse kinematic analysis of the robotic arm are completed, and the working space of the four-axis robotic arm used in this paper is determined. At the same time, the joint space and Cartesian space trajectory planning are compared to determine the algorithm used for the path planning of the robotic arm, which provides a theoretical basis for the construction of the robotic arm's motion control. In addition, the mechanical structure design and the control system design are implemented to verify the proposed kinematic theory and to lay the foundation for the subsequent performance experiments. Furthermore, the system introduces improved target detection and YOLOV7 visual recognition algorithm to improve the recognition speed and accuracy of the algorithm deployed on embedded devices. It is experimentally verified that the proposed robotic arm system achieves an accuracy of 95.2% when reaching a 29FPS frame rate, which is better than traditional target detection algorithms such as Faster-RCNN, SSD, and YOLOV5. Based on the realization of the basic motion, an upper computer based on Python language and QT is built into the embedded device Jetson Nano, which includes motion control, target recognition, and serial communication.

**Figure 1.** The system architecture of the robotic arm system.

Aiming at the problems of high cost, low recognition accuracy, and difficult design of traditional manipulators, this paper designs and improves a lightweight four-degree-of-freedom manipulator. The manipulator is based on the YOLOv7 target recognition algorithm and depth perception algorithm. The arm adds visual guidance. The paper mainly has the following innovations:

(i). Introducing embedded devices into the robotic arm system and building a deep learning algorithm on top of it improves the intelligence of the robotic arm and greatly reduces the cost of the robotic arm.

(ii). Combining the deep learning algorithm with the depth camera, the system can quickly obtain the multi-dimensional information of the target.

(iii). Based on the QT platform, a robotic arm control host computer is built, which makes the system portable and highly expandable.

A comparison between our robotic arm system and some recent robotic arm designs is given in Table 1. It can be seen from Table 1 that the intelligent robotic arm system

designed in this paper is complete compared to the four robotic arm systems. In addition, the detection accuracy of our design is also better than the other designs. Compared with the traditional industrialized system design, the robotic arm has low power consumption and high portability due to the use of the embedded system.

**Table 1.** Comparison of the proposed robotic arm system and related designs.

| Ref. | Robotic Arm Simulation | Vision System | Target Recognition Algorithm | Upper Computer | Embedded Systems | Detection Accuracy |
|---|---|---|---|---|---|---|
| [9] | Yes | Yes | No | Yes | No | No |
| [10] | Yes | Yes | Yes | No | No | 91.67% |
| [11] | Yes | Yes | No | Yes | No | No |
| [12] | Yes | Yes | Yes | No | No | 63% |
| [13] | Yes | Yes | Yes | No | No | No |
| [14] | Yes | Yes | Yes | Yes | No | No |
| This work | Yes | Yes | Yes | Yes | Yes | 95.2% |

The motion algorithm of the robotic arm body includes the basic motion algorithm and the trajectory planning algorithm. After the control algorithm is determined, the hardware of the robotic arm control system is designed, including the construction of the control system platform and the corresponding driver code implementation, along with the production of the physical robotic arm using 3D printing and related technologies. Once the robotic arm hardware platform is perfected, the target recognition and visual guidance algorithms, as well as the upper computer visualization software, need to be built.

The research purposes of this article are as follows:

- Carry out simulation experiments based on the kinematics theory of the manipulator, complete the modeling of the manipulator and the forward and reverse kinematics analysis, and determine the working space of the four-axis manipulator used, laying the foundation for trajectory planning and motion control;
- Using joint space and Cartesian space trajectory planning, determine the algorithm used by the robot arm for path planning, which provides a theoretical basis for the construction of robot arm motion control;
- Carry out mechanical structure design and control system design for the robotic arm, and complete the physical design and realization of the robotic arm, thereby verifying the proposed kinematics theory and laying the foundation for subsequent performance experiments;
- Adopt the improved target detection and YOLOV7 visual recognition algorithm to improve the recognition speed and accuracy of the algorithm deployed on the embedded device, effectively improve the performance of the robotic arm, make the system portable and highly expandable, and achieve the goal of the project's expected requirements.

The research content of this article includes:

1. Carrying out kinematics analysis and design, mainly including modeling the whole machine, performing forward and reverse kinematics analysis, and introducing improved target detection and the YOLOV7 visual recognition algorithm;
2. Structural analysis and design of the manipulator, design, and manufacture of the physical model of the manipulator, the use of 3D printing and related technologies to make the real manipulator, and conducting forward and inverse kinematics analysis;
3. Designing the control system of the manipulator and constructing the upper computer of the control system based on Python language and QT in the embedded device Jetson Nano;
4. Deploying the target detection algorithm, mainly including repeated experiments to collect relevant data and make a dataset, training the robot arm through the algorithm, and analyzing, optimizing, and improve it.

5. Carrying out the software design of the upper computer, mainly including the design of the basic interface used by the user and embedding the relevant control algorithms involved.

## 2. Kinematic Analysis and Design of Robotic Arms

As the basis for the motion control of the robotic arm, the kinematic analysis of the robotic arm is the study of the relationship between the motion of the robotic arm in each coordinate system [15,16]. Before carrying out the kinematic analysis of the robotic arm, the arm is first modeled in Matlab using the D-H model with the following basic parameters: theta is the joint angle, d is the linkage offset [17,18], a is the linkage length, and alpha is the linkage torsion angle.

The basic joint model can be found in Figure 2. Once the model has been established, the workspace of the joint model can be determined by the Monte Carlo method [19,20]. In this section, the Monte Carlo method is used to solve the workspace as follows [21]: firstly, random variables are generated for each joint, and a random set of joint space vectors are generated for the robotic arm, which is calculated by using 10,000 points. Secondly, the kinematic positive solution is calculated and mapped from the joint space to the end workspace (Cartesian coordinate system), and finally, the result is plotted in Figure 3. It can be seen from Figure 3 that the result of the arm motion contains any coordinate in the 3D space, which shows that the designed robotic arm can reach any specified point in the actual work. After the arm has been modeled, a kinematic analysis of the arm is carried out, including both positive and negative kinematics.

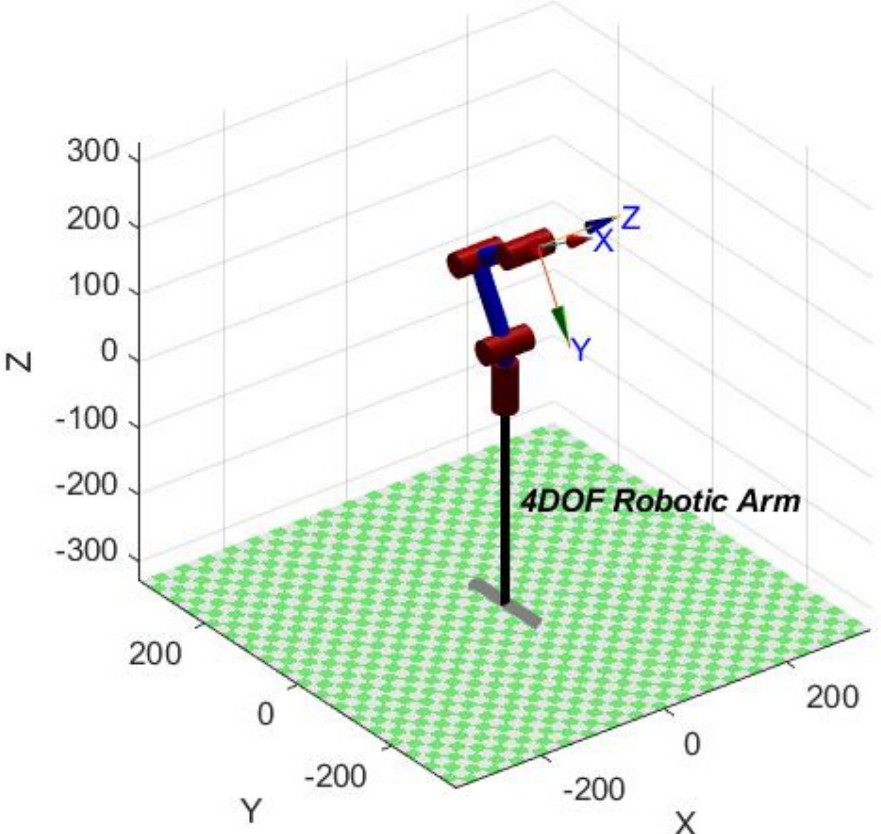

**Figure 2.** Robotic arm joint model.

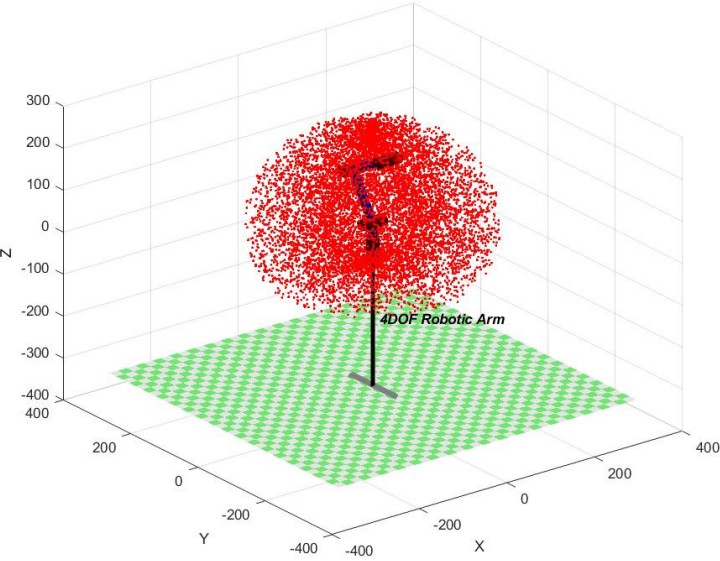

**Figure 3.** Robotic arm workspace.

Figure 4 reflects the movement space that the manipulator can reach in different coordinate systems

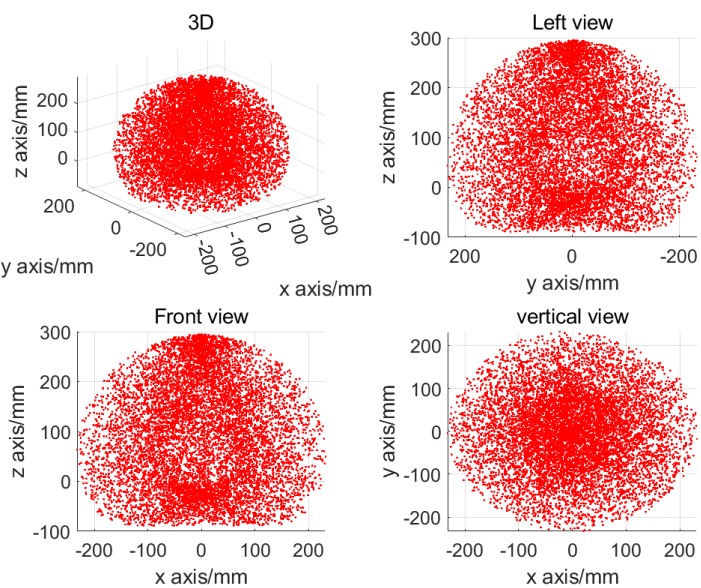

**Figure 4.** The movement space of the robotic arm.

### 2.1. Positive Kinematic Analysis of the Robotic Arm

Positive kinematic analysis of the robotic arm refers to obtaining the position of the end of the robotic arm relative to the reference coordinate system based on these angles and information about the connecting rod [22], given that the rotation angles of the motor at each joint between the robotic arms are known. The structure of the four-axis robotic arm in this paper is shown in Figure 5 where Joint0 is the base coordinate system of the robotic arm, Joint1, Joint2, and Joint3 are all rotational joints, and Joint4 is the wrist joint.

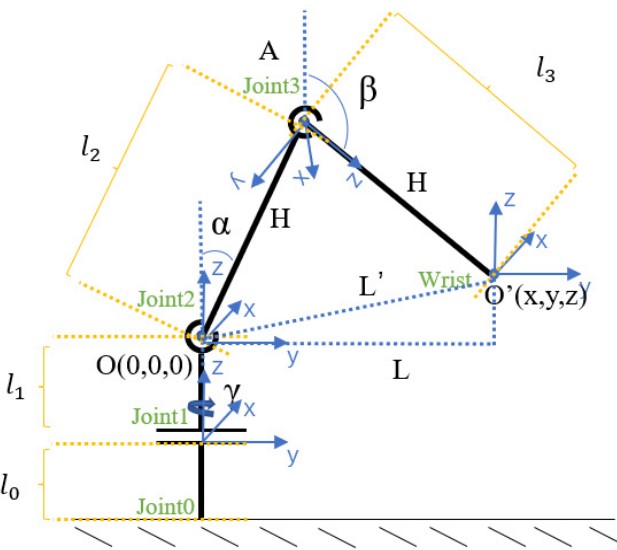

**Figure 5.** The schematic diagram for positive kinematic analysis of the robotic arm.

The following equation for the positive kinematics of a 4-degree-of-freedom robotic arm is derived. Firstly, according to the transformation matrix, the linkage can be defined as:

$$
{}^1_0T = R_Z(\gamma) = \begin{bmatrix} \cos(\gamma) & \sin(\gamma) & 0 & 0 \\ \sin(\gamma) & \cos(\gamma) & 0 & 0 \\ 0 & 0 & 0 & 0 \\ 0 & 0 & 0 & 1 \end{bmatrix} \tag{1}
$$

$$
{}^1_2T = R_x(90°) * R_Z(\alpha) = \begin{bmatrix} \cos(\alpha) & -\sin(\alpha) & 0 & 0 \\ 0 & 0 & 1 & 0 \\ -\sin(\alpha) & -\cos(\alpha) & 0 & 0 \\ 0 & 0 & 0 & 1 \end{bmatrix} \tag{2}
$$

$$
{}^2_3T = D_x(l_2) * R_Z(\beta) = \begin{bmatrix} \cos(\beta) & -\sin(\beta) & 0 & l_2 \\ \sin(\beta) & \cos(\beta) & 0 & 0 \\ 0 & 0 & 0 & 0 \\ 0 & 0 & 0 & 1 \end{bmatrix} \tag{3}
$$

$$
{}^3_4T = D_x(l_3) = {}^2_3T = D_x(l_2) * R_Z(\beta) = \begin{bmatrix} 1 & 0 & 0 & l_3 \\ 0 & 0 & 0 & 0 \\ 0 & 0 & 0 & 0 \\ 0 & 0 & 0 & 1 \end{bmatrix} \tag{4}
$$

Based on the already defined linkage coordinate system and the corresponding linkage parameters, the transformation matrix of the tool coordinate system with respect to the base coordinate system can be derived from the following equation:

$$
{}^0_4T = {}^0_1T * {}^1_2T * {}^2_3T * {}^3_4T \tag{5}
$$

Transforming the above equation yields the following equation:

$$
{}^0_4T = R_Z(\gamma) * R_x(90°) * R_Z(\alpha) * D_x(l_2) * D_x(l_2) * R_Z(\beta) * D_x(l_3) \tag{6}
$$

Expanding the above equation yields:

$$
{}^0_4T = \begin{bmatrix} c_1 * c_{23} & -c_1 * s_{23} & 0 & c_1 * (l_3 * c_{23} + l_2 * c_2) \\ s_1 * c_{23} & -s_1 * s_{23} & c_1 & s_1 * (l_3 * c_{23} + l_2 * c_2) \\ s_{23} & c_{23} & 0 & l_3 * s_{23} + l_2 * s_2 \\ 0 & 0 & 0 & 1 \end{bmatrix} \tag{7}
$$

where $c_1$, $c_2$ is shorthand for $\cos(\theta_1)$, $\cos(\theta_2)$
$s_1$, $s_2$ is shorthand for $\sin(\theta_1)$, $\sin(\theta_2)$,
$c_{23}$ is shorthand for $\cos(\theta_2 + \theta_3)$,
and $s_{23}$ is shorthand for $\sin(\theta_2 + \theta_3)$.

From the fourth column of the ${}^0_4T$ matrix, the position of the wrist joint at the base of the robotic arm Joint0 can be obtained:

$$
x = \cos(\gamma) * (l_2 * \cos(\alpha)) + l_3 * \cos(\alpha + \beta) \tag{8}
$$

$$
y = \sin(\gamma) * (l_2 * \cos(\alpha)) + l_3 * \cos(\alpha + \beta) \tag{9}
$$

$$
z = -l_2 * \sin(\alpha) + l_3 * \cos(\alpha + \beta) \tag{10}
$$

According to the above equation, the three-dimensional coordinates of the end of the robotic arm in space can be calculated once the rotation angle $\alpha$, $\beta$ and the length of each joint of the robotic arm are known.

*2.2. Robotic Arm Inverse Kinematics Analysis*

In contrast to the positive kinematic analysis of the arm, the inverse kinematic analysis of the arm uses calculations to solve for the angle of rotation at each joint when the relevant parameters of the connecting rod and the position of the end of the arm concerning the reference coordinate system have been obtained [23]. Geometric analysis based on the structure of the robotic arm: for the robotic arm involved in this paper, it is necessary to find the angle of rotation at each joint when the end of the robotic arm is moved from O to O′, i.e., to find $\gamma$, $\alpha$, and $\beta$, according to the inverse kinematics requirements.

Firstly, the geometric relationship between the rotation angle of the base and the coordinates of the end of the arm is analyzed, i.e., $\gamma$.

Assuming that the base coordinate system is O, the end coordinate system is O′, the coordinates are (x, y, z), and the arm length is H. When the rotation angle of the base of the arm is $\gamma$, the following equation can be obtained according to the Pythagorean theorem:

$$
L = \sqrt{x^2 + y^2} \tag{11}
$$

where $L$ is the projection in the xoy plane of the coordinate system O′ at the end of the robotic arm concerning the base coordinate system O. The following equation is obtained from the sine theorem:

$$
\gamma = \sin^{-1}\left(\frac{x}{L}\right) \tag{12}
$$

where $\gamma$ is the angle at which the base of the arm needs to be rotated relative to the base coordinate system when the end of the arm moves to O′.

Next, we will discuss $\alpha$. There are two cases for $\alpha$, as shown in Figure 6.

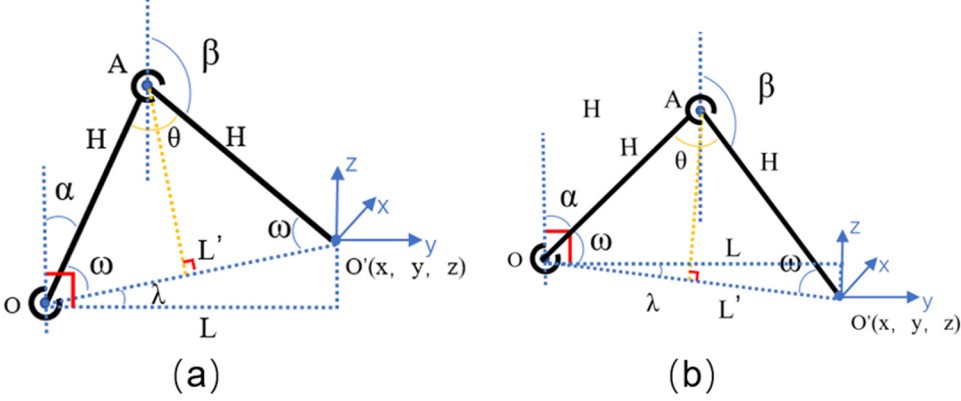

**Figure 6.** Angle diagram of the robotic arm: (**a**) $z > 0$, (**b**) $z < 0$.

When $z > 0$, according to the Pythagorean theorem, we can obtain:

$$L' = \sqrt{L^2 + z^2} \tag{13}$$

Since the lengths of the two arms OA and O′A of the robotic arm are the same and known, the following equation can be driven according to trigonometric relationship:

$$\theta = 2 * \cos^{-1}\left(\frac{L'}{2 * H}\right) \tag{14}$$

And:

$$\omega = \left(\frac{\pi}{2} - \frac{\theta}{2}\right) \tag{15}$$

$$\lambda = \cos^{-1}\left(\frac{L}{L'}\right) \tag{16}$$

where $\omega$ is the angle formed by the robotic arm OA and $L'$ in space and $\lambda$ is the angle formed by $L$ and $L'$ in space. Therefore, it can be obtained that, at $z > 0$, when the end of the robotic arm moves to O′, the angle $\alpha$ that the robotic arm polar coordinates need to be rotated is

$$\alpha = \left(\frac{\pi}{2} - \lambda - \omega\right) \tag{17}$$

The final formula for calculating the alpha angle is

$$\alpha = \left(\cos^{-1}\left(\frac{\sqrt{x^2 + y^2 + z^2}}{2 * H}\right) - \cos^{-1}\left(\frac{\sqrt{x^2 + y^2}}{\sqrt{x^2 + y^2 + z^2}}\right)\right) \tag{18}$$

When $z < 0$, according to the Pythagorean theorem and Equations (5) and (6):

$$\alpha = \left(\frac{\pi}{2} + \lambda - \omega\right) \tag{19}$$

The formula for calculating the $\alpha$ angle is obtained as:

$$\alpha = \left(\cos^{-1}\left(\frac{\sqrt{x^2 + y^2 + z^2}}{2 * H}\right) + \cos^{-1}\left(\frac{\sqrt{x^2 + y^2}}{\sqrt{x^2 + y^2 + z^2}}\right)\right) \tag{20}$$

The final beta angle is easily obtained as:

$$\beta = \pi + \alpha - \theta \tag{21}$$

The above is the process of finding the inverse motion of the robotic arm based on the geometric method. The conversion from spatial coordinates to the rotation angle of the robotic arm motor can be completed by converting the above formula into code and embedding it in the lower computer.

### 2.3. Robotic Arm Path Planning Analysis

After analyzing the forward and reverse kinematics of the robotic arm, to ensure that the trajectory of the robotic arm can meet the design requirements of fast response, low motion inertia, and easy control when working, the joint space trajectory planning and Cartesian space trajectory planning for the model are carried out for a robotic arm reaching the one point from another point in space using two algorithms, respectively, as shown in Figures 7 and 8.

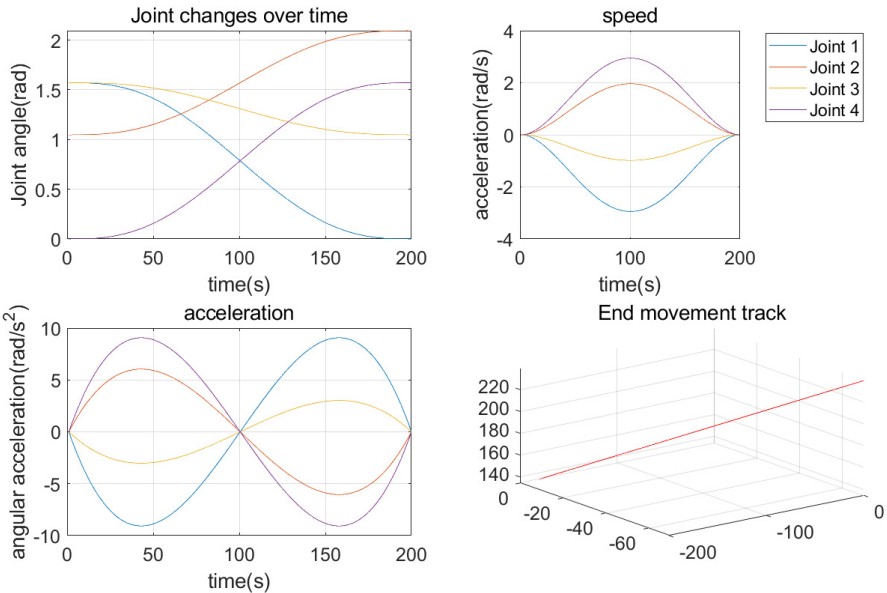

**Figure 7.** Cartesian trajectory planning for robotic arms.

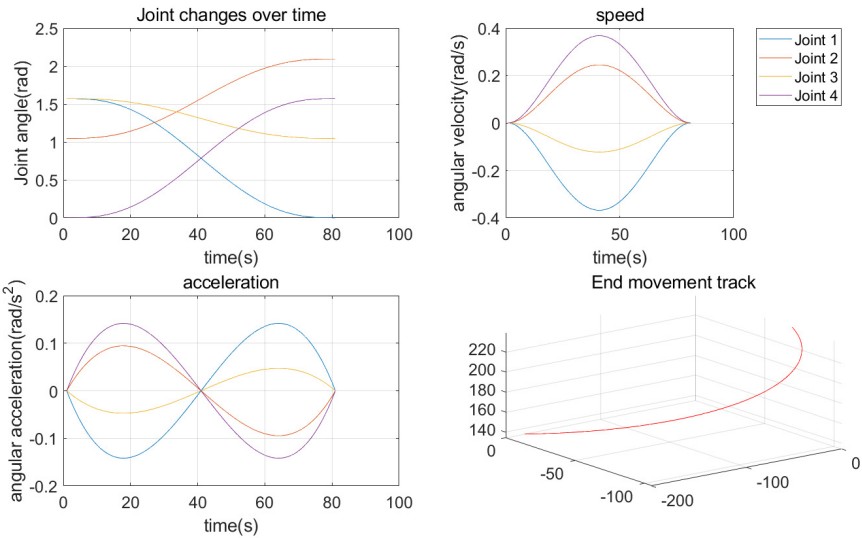

**Figure 8.** Spatial trajectory planning of robotic arm joints.

From comparing the joint time variation in Figures 6 and 7 we can obtain that joint space trajectory planning takes less time than Cartesian trajectory planning with the same starting point, and its end is circular, which makes it more suitable for places with strict

time response requirements. So, the joint space trajectory planning algorithm is used in the proposed system due to the system being based on an embedded device and requiring real-time control.

## 3. Vision-Guided Design of the Four-Axis Robotic Arm System

When designing the mechanical structure of a robotic arm, the economy of production and processing, as well as the practicality in the use of scenarios, should first be considered [24]. It is often necessary to design the corresponding robotic arm structure for different tasks in a specific scenario. Firstly, the 3D model was constructed and assembled using Autodesk Inventor software, and engineering drawings were carried out. The main structure of the robotic arm consists of three main axes and joints: the base, the arm, and the end-effector. The end-effector was selected as the gripper type. A four-bar mechanism is used to build the arm body. And the main transmission method of the arm is the belt drive, with the idler pulley above the stepper motor connected to the gears by belts at the three joints, and each joint is controlled by an independent motor.

The main structure was printed out using 3D printing technology, after which the main components were assembled and commissioned. The robotic arm mainly uses pulleys and idler pulleys instead of the generation of gears, which greatly improves the stability of the system. The design of the limit switch mounting holes and the corresponding wire collation module was carried out so that the signal wires do not interfere with the normal movement of the motor. The base is fixed using bearings and M6 screws + bearings to ensure the stability of the base. After testing, the robotic arm can complete the basic movement requirements, i.e., zeroing and moving to the corresponding position. The final result is shown in Figure 9. After the body of the arm has been installed and assembled, the camera mounting platform and the controller mounting platform are designed and printed out to complete the total assembly.

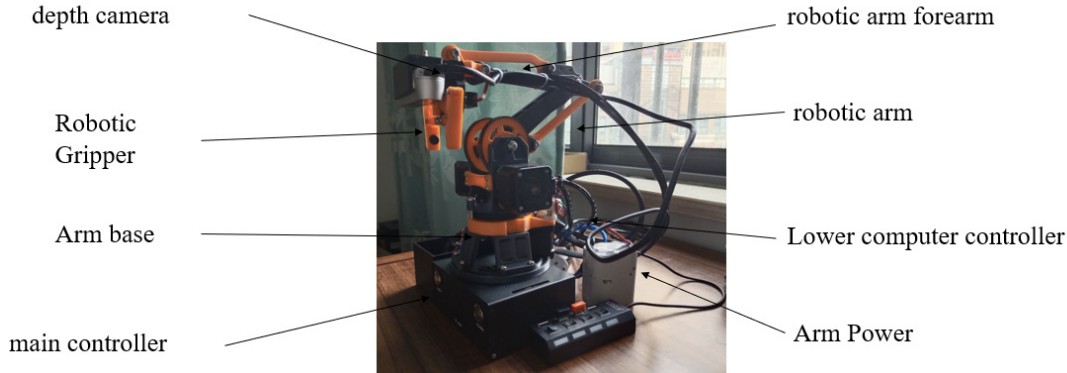

**Figure 9.** The physical picture of the robotic arm.

### 3.1. Robotic Arm Control System Design

3.1.1. Upper Computer Implementation

Due to the need to use the deep learning algorithm Yolov7, the Jetson Nano is chosen as the host controller for this paper after comprehensively considering the computing power and code running effect. The upper controller is externally connected to the camera and the robotic arm, and the software algorithm is embedded internally to implement the image recognition function and the robotic arm control function. Furthermore, because of the depth data of the target in the image being used, the depth camera RealSense D435i from Intel is chosen, which integrates two IR stereo cameras, one IR projector, and one color camera.

To control the work of the lower computer, the upper computer software based on the QT software development platform is compiled using the Python language. PYQT is the implementation of the QT library in Python, which is a convenient interface for Python

programming and thus enables graphical interface programming. The basic interactive interface was designed in conjunction with the actual requirements of the robotic arm control system by taking into account factors such as efficient collaboration between software subsystems, reasonable interface layout, user-friendly interaction, and user habits, as shown in Figure 10.

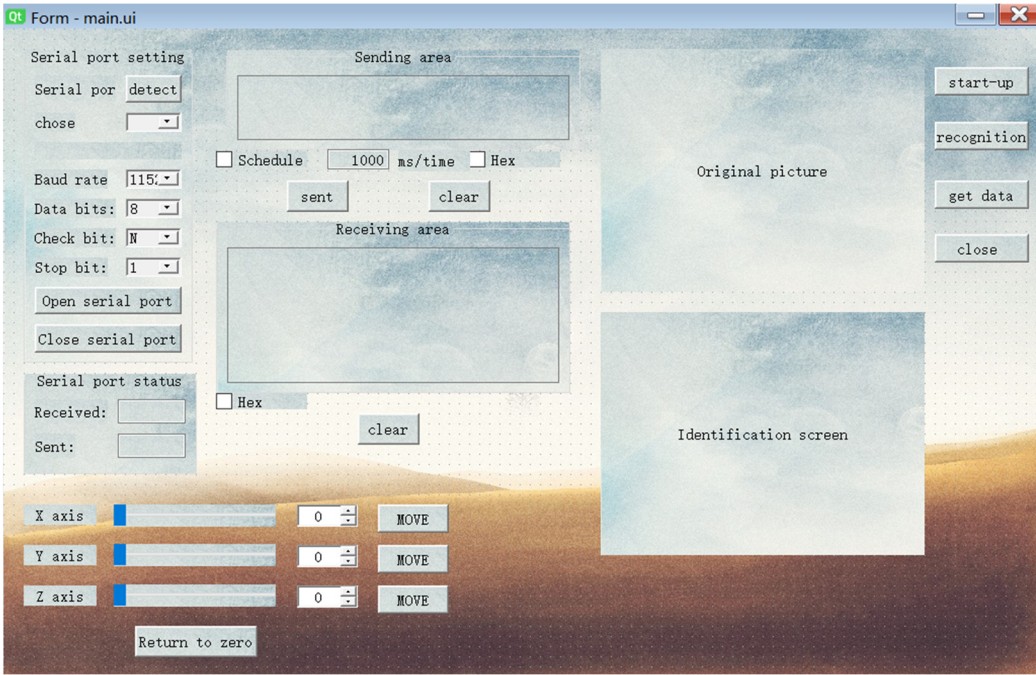

**Figure 10.** Upper computer interface diagram.

The main interface contains the following functional areas: serial information area, identification interface display area, and motor control area. The following are the roles of these three sections:

1. Serial port connection display area: The design goal of this area is to facilitate the user to use the serial communication method to connect to the lower Arduino controller through the upper computer software. This area can be set up with a series of parameters such as baud rate, serial number, etc. There are functions such as serial port detection, serial port setting and sending motion information, and receiving answer signals from the lower computer.
2. Recognition interface display area: The recognition interface display area is mainly used to show the recognition situation of the camera, and also to allow the user to directly see the position information of the target in the camera.
3. Sliding bar control motor area: To facilitate the user to operate the robotic arm directly, this area is set up so that the user can control the movement distance of the motor by clicking on the various data in the sliding bar or directly by entering the numbers after the sliding bar.

Next, the YOLOV7 target detection algorithm and the depth detection algorithm are embedded in the software. For the YOLOV7 target detection algorithm, a configuration file is first loaded in the constructor along with randomly generated colors for each category, and the model is initialized. To improve processing efficiency, the image method is first preprocessed, i.e., the array is converted from a discontinuous array in memory to a continuous array in memory, making it run faster. After the pre-processing, the detect method calls the processed array and performs image normalization followed by model inference and NMS non-maximum suppression of the inferred results. Finally, a rectangular box with labels is drawn when a target object is present in the image

For the depth detection algorithm, the different images of the open camera and the return values are mainly defined. The depth image of the camera is first configured in the constructor, after which the various internal parameters of the camera can be obtained via the library functions of Realsense. Once the parameters have been obtained, the camera parameters, depth parameters, color map, depth map, etc., are returned.

3.1.2. Lower Computer Design

The main functions of the lower computer in the system are:

1. Communicating with the upper computer through the serial port;
2. Carrying out the inverse operation of the robotic arm to decode the coordinate information sent back from the upper computer into the level signal of the motor control pins to control the movement of the robotic arm to the specified position;
3. Controlling the servo pins to achieve the opening and closing movement of the hand claw, to collect the limit switch signal, and to control the robotic arm to zero.

Based on the above functional requirements, the lower computer circuit we designed is mainly composed of Arduino MEGA2560+RAMPS1.4+A4988+stepper. Arduino Mega 2560 is an upgraded version of the Arduino MEGA series. The control board is equipped with an ATmega2560 chip and a 16 M Hz crystal oscillator. There are 54 digital IN/OUT ports, and among these pin ports, 15 can be pulse width modulated (PWM) and output; there are 16 analog signal ports; 4 serial input and output ports; a USB port; and a reset button, etc. Its power supply mode can be a USB power supply or DC port direct input. Its programming method uses the compilation environment that comes with Arduino to realize code writing.

A Stepper motor is an electromagnetic actuator that can convert the input digital signal into the rotary motion of its central axis. It has the advantages of easy control, adjustable speed, high motion precision, small motion inertia, and less accumulated error. Widely used in electromechanical integration equipment, the rotation angle of the central axis depends on the number of input pulses, which is positively correlated. At the same time, the rotation speed of the stepper motor can be controlled by changing the input pulse frequency. When selecting and using a stepper motor, it is necessary to consider the speed, torque, and no-load starting frequency:

(1) When selecting a stepper motor, its speed is an important referenced factor because the speed of the motor determines the output torque of the motor.
(2) The choice of the torque of the stepping motor. The torque of the stepping motor is mainly determined for the work tasks in different scenarios. Generally speaking, motors with smaller shaft diameters such as 20, 28, and 42 are usually used. In the torque scenario (below 0.8 N·m), 57 stepper motors are more suitable for a medium torque (about 1 N·m), and for larger torques, stepper motors with larger shaft diameters such as 86, 110 should be selected.

Given the above stepper motor selection indicators, the project finally decided to use three 2-phase, 4-wire, 42-stepper motors as the power source of the system. The important indicators are in Table 2.

**Table 2.** The main index parameters of the motor.

| Motor Model | Body Length (mm) | Step Angle (°) | Number of Phases (NO.) | Rated Voltage (V) | Rated Current (A) | Moment of Inertia (g cm$^3$) |
| --- | --- | --- | --- | --- | --- | --- |
| B0459 | 47 | 1.8 | 2 | 2.6 | 2.0 | 82 |

In the case that the stepper motor has been confirmed to choose a 42-stepper motor, we select the driver based on the following points:

1. Determine the torque required for the load carried by the motor, and the torque of the selected driver must meet the requirements of the motor.

2. Determine the rated current of the stepping motor used. Generally, the output current of the driver can be set within a certain range. If the current is too small, the output torque of the motor may be insufficient. The rated current used in this article is 2.0 A.

3. Determine the working voltage of the motor. The output voltage of the driver must be within the range of the motor. The voltage used in this article is 2.6 V.

4. The number of phases must be equal, the motor is two-phase, the step angle is 1.8°, and the driver must be two-phase.

The main function of the motor driver is to regulate the subdivision and control the motor. Ramps are powerful Arduino expansion boards with multiple functional interfaces such as a 12,864 screen interface, 5 stepper motor interfaces, temperature control, limit switch interfaces, etc. Its circuit schematic diagram is shown in Figure 11.

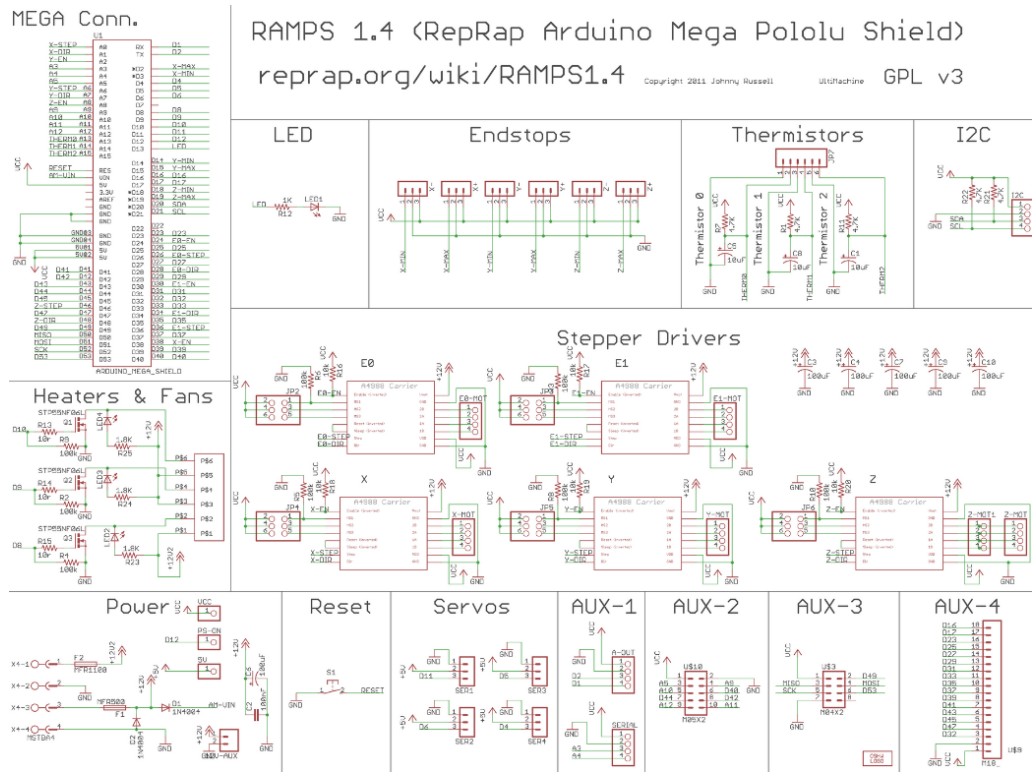

**Figure 11.** Schematic diagram of ramps.

A4988 motor driver is a common drive module in CNC technology and mechatronics. It has the characteristics of an affordable price, easy control, small size, and rich subdivision. The wiring diagram of the stepper motor driver A4988 is shown in Figure 12.

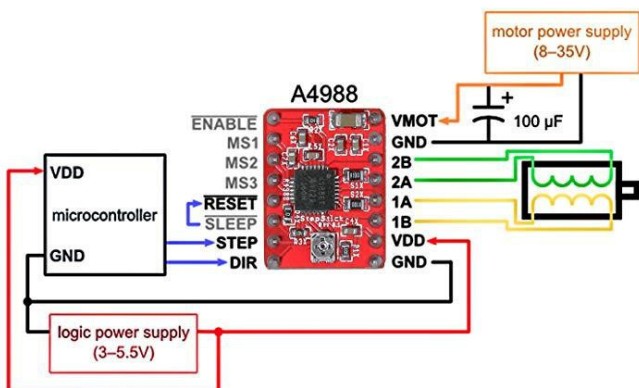

**Figure 12.** A4988 pin definition map.

The underlying code implementation mainly includes the communication program, the motion solver layer, and the motor motion control layer.

The main way for the upper computer to control the work of the lower computer is to use serial communication to send different data. After the upper computer is connected to the lower computer, the corresponding baud rate and serial port number on the upper computer side need to be selected, and Open Serial Port is clicked to open the serial port.

For example, to control the three-axis zeroing, the upper computer sends the "H" character when the serial communication is enabled and the communication is stable, and the lower computer will settle the character and match it to the corresponding action after receiving it. For the convenience of users, the specific communication method will be displayed in the upper computer receiving area after this connection, and users can follow the corresponding example to control the movement of the robotic arm.

After the upper computer sends the corresponding 3-dimensional coordinate points to the lower computer, the motion solver layer program realizes the motor motion inverse solution using kinematic analysis to calculate the distance that the corresponding motor needs to move. The motor motion control layer is used to realize the specific motor motion control. To achieve program reuse and easy maintenance of the code, the underlying driver code is written and declared in the corresponding file using the C++ programming language and instantiated in the main file. For the motor control program, firstly, the motor class is defined, and its member variables and properties are declared, including motor pins, speed, etc. After that, functions are defined to control the movement distance of the motor by specifying the number of pulses.

### 3.2. Vision Control Algorithms

As the most typical representative of one-stage target detection algorithms [25], the YOLO algorithm can be used in real-time systems due to the fast running time, which is based on deep neural networks for object recognition and localization. To ensure that high-speed detection is possible on embedded platforms, the YOLOV7 algorithm is used in this paper, which is the most advanced algorithm in the current YOLO series and surpasses the previous YOLO series in terms of accuracy and speed [26,27]. In addition, in this paper, we use the TensorRT optimizer, a deep learning inference engine introduced by NVIDIA [28,29], which can accelerate the model during inference and improve the recognition speed of the model for images as well as the read-in and read-out efficiency [30,31]. Currently, the model can be applied in common deep-learning frameworks [32].

After the deployment method is determined, the dataset is produced with the robotic arm recognizing and grasping objects, mainly including 8 common objects such as cell phones, glasses, tea cups, and Bluetooth headphones. Firstly, a total of 3 datasets were made to train the model for the pictures of the photographed objects, but the final trained model had poor anti-interference ability and low detection accuracy due to the single background of the photographed objects. After several attempts, a model with better detection accuracy was trained by increasing the background of the images, the number of images, and the number of training sessions. After the dataset was created, the target annotation was labeled. After all the images in the dataset are labeled, the overall dataset is created, and the original dataset is classified according to a certain ratio. In this paper, a ratio of 7:3 is used to divide the dataset into a training dataset and a test dataset.

After the dataset is divided, the images in the dataset need to be trained. First, YOLOv7 is downloaded, and data inference is performed on a cloud-based server to ensure the inference rate. The parameters of the server are pytorch1.10, python3.8, CUDA11.3, RTXA4000 GPU, 16 GB video memory, 12-core Intel(R) Xeon(R) Gold 5320 CPU @ 2.20 GHz, and 32 GB memory. According to the above server configuration, the corresponding training parameters are set. The main parameters include input image size ($640 \times 640$), number of training iterations (200), batch size (16), learning rate (0.01), and momentum (0.937). The changes in various values during the training process as the number of iterations increases is shown in Figure 13. It can be seen from Figure 13 that after 200 training iterations, the

values of high precision and recall do not fluctuate greatly, and the changes in each value tend to be smooth, indicating an excellent training result.

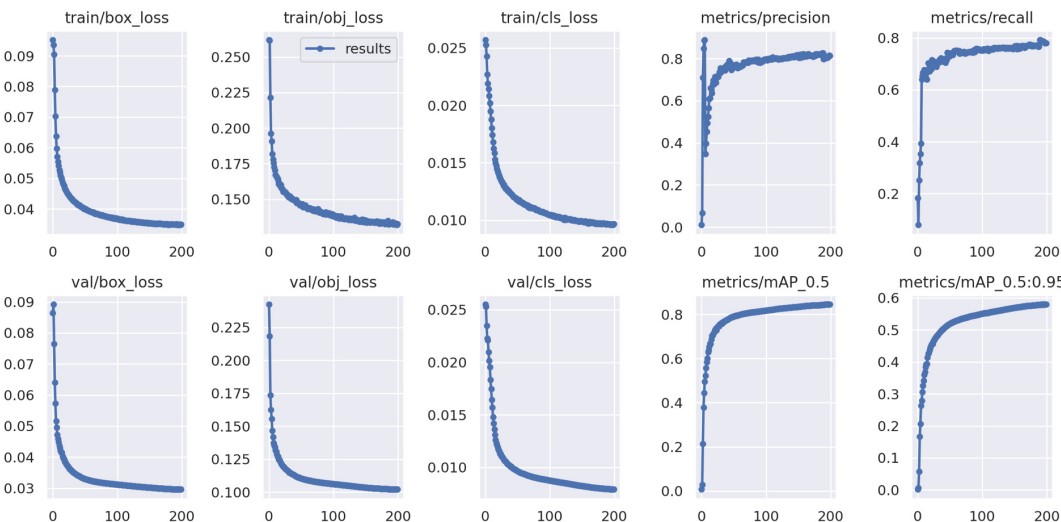

**Figure 13.** Training results of the robotic arm system.

## 4. Experimental Results and Discussion

After the various parts of the robotic arm system have been designed, to verify the accuracy performance of the proposed system as well as the recognition performance, the end repetition accuracy of the robotic arm and the results of the robotic arm target detection algorithm are subsequently tested.

### 4.1. Robotic Arm End Repetition Accuracy Test

The international standard ISO 9283-1998 was used to evaluate the performance of robotic arms [33], focusing on the end repetition accuracy of robotic arms. End repetition accuracy is an important index describing the motion performance of the whole robotic arm, which is closely related to the stiffness of the hardware structure of the robotic arm body, transmission errors, and the corresponding level of motion control [34]. Figure 14 shows the position repetition accuracy described in the ISO 9283 standard, where the value of RPl is the required position repetition accuracy.

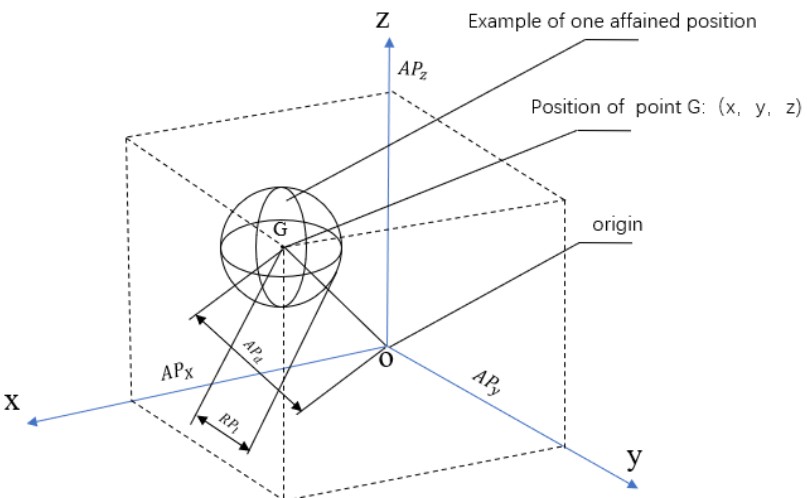

**Figure 14.** Position repeatability diagram.

The size of the sphere at all arrival position points of this envelope is defined in a mathematical sense according to the formula for RPl, as defined by ISO 9283 [35]:

$$RP_l = L^1 + 3S_l \tag{22}$$

where $L^1$ is the average of the distances of all arrival positions from their center of gravity, calculated as:

$$L^1 = \frac{1}{n}\sum_{j=1}^{n} l_j \tag{23}$$

$$l_j = \sqrt{\left(x_j - x^1\right)^2 + \left(y_j - y^1\right)^2 + \left(z_j - z^1\right)^2} \tag{24}$$

where $x_j$, $y_j$, $z_j$ are the spatial Cartesian coordinates of each arrival position; $x^1$, $y^1$, $z^1$ are the spatial Cartesian coordinates of the center of gravity of each arrival position; and $S_l$ is a standard deviation:

$$S_l = \sqrt{\frac{\sum_{j=1}^{n}\left(l_j - l^1\right)^2}{n-1}} \tag{25}$$

The practical physical meaning of the calculation using the 3θ principle is that a specified amount of standard deviation is added to the average value, resulting in a minimum value that encompasses the diameter of all spheres reaching the position point, i.e., the position repeatability accuracy. The actual end repeatability of the robotic arm in this paper was 0.05 mm when tested at a rated speed of the stepper motor under a load of 1 kg.

### 4.2. Robotic Arm Target Detection Algorithm Testing

To demonstrate the effectiveness of yolov7 deployed on the embedded platform Jetson Nano, the same dataset was tested on the same device using different algorithms. Figure 15 shows the effect of target recognition, using a mouse as an example.

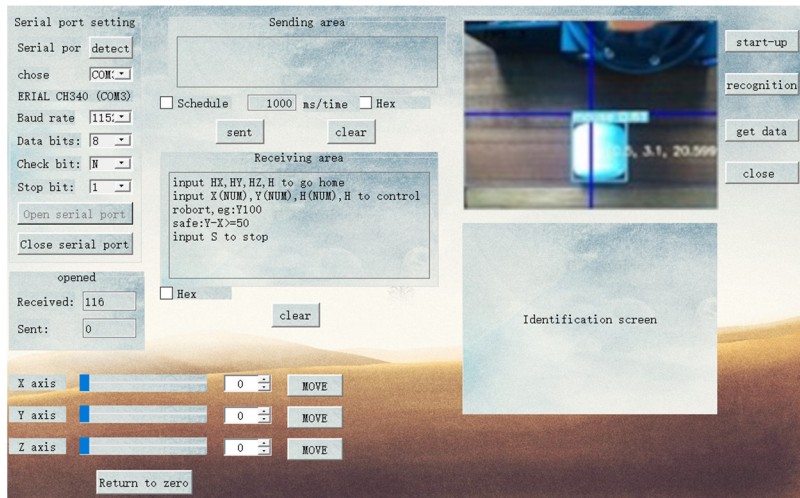

**Figure 15.** Object recognition effect.

The main algorithms compared are the YOLOV5 [36], Faster-RCNN [37], SSD [38], and YOLOV7 [39], the results of which are shown in Table 3. It can be seen that the average detection rate as well as the speed of yolov7 outperformed the other algorithms, indicating that the YOLOV7 algorithm is more suitable for deployment on embedded platforms than other algorithms.

**Table 3.** Comparison between different algorithms.

| Algorithm | Detection Accuracy (%) | | | Average Accuracy (%) | Speed/(fps) |
|---|---|---|---|---|---|
| | Mobile Phones | Teacups | Eyeglasses | | |
| Faster RCNN | 88.6 | 89.2 | 91.3 | 89.7 | 1 |
| SSD | 88.9 | 88.6 | 89.2 | 88.9 | 8 |
| YOLOV5 | 91.9 | 91.7 | 92.7 | 92.1 | 20 |
| YOLOV7 | 94.8 | 95.2 | 95.6 | 95.2 | 29 |

Next, to evaluate the power consumption performance of the robotic arm, under the same working conditions, we use the rotation center of the big arm as the system origin (0, 0, 0), and make the end of the robotic arm move from point (100, 100, 0) to point (200, 100, 0) at a movement speed set to 200 mm/s and a set load of 1000 N and compare the driving power of the three robotic arms, as shown in Figure 16. It is obvious that the driving power of the robotic arm designed in this paper is less than the other two robotic arms, so our robotic arm has the feature of low power consumption.

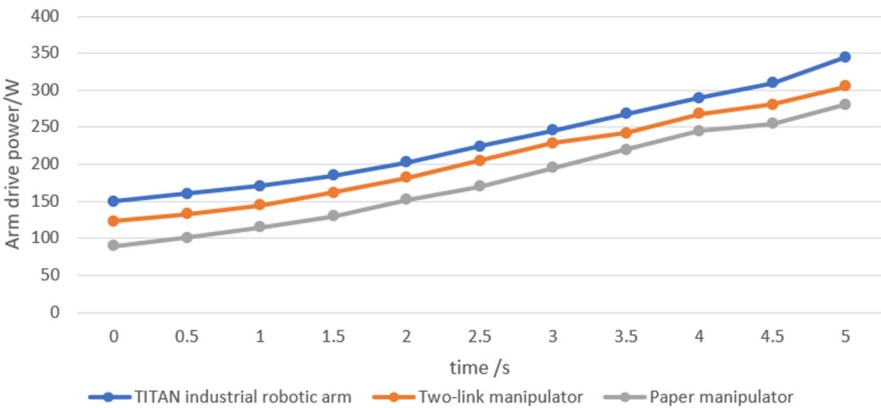

**Figure 16.** Comparison results of three kinds of robotic arm drive power.

## 5. Conclusions

Aiming at the problems of the high cost, low recognition accuracy, and difficult design of traditional manipulators, this paper designs and improves a lightweight four-degree-of-freedom manipulator. The manipulator is based on the YOLOv7 target recognition algorithm and depth perception algorithm. The arm adds visual guidance. To reduce the cost of the robotic arm, this paper introduces embedded devices into the robotic arm system and builds a deep learning algorithm on top of it to improve the intelligence of the robotic arm and obtain multi-dimensional information about the target. In terms of mechanical movement accuracy, the mechanical arm designed in this paper can achieve an end accuracy of 0.05 mm under the condition of a load of 1 kg using the ISO 9283 international standard. The lower recognition accuracy reaches 95.2%. At the same time, after relevant experiments, the overall power consumption of the manipulator is also lower than that of the traditional manipulator. Finally, based on the QT platform, a robotic arm control host computer is built, which makes the system portable and highly expandable. After relevant experimental verification, the results show that applying the object recognition and grasping functions to the four-axis robotic arm can greatly improve the intelligence of the robotic arm, while greatly reducing the robotic arm's design and manufacturing costs and power. At the same time, this system has the characteristics of strong portability, so it can be considered to be applied to the six-axis robot arm. More work tasks can be realized on the robotic arm. Therefore, this paper is conducive to promoting the wide application of manipulators in industrial environments and improving the construction of industrial automation. It is

worth mentioning that the robotic arm designed in this paper has the following directions for further improvement:

(i) Continue to realize the optimization processing of the manipulator recognition algorithm and improve the deep reinforcement learning ability of the manipulator;

(ii) Manipulators with different structures/dimensions should be produced to meet actual application requirements.

**Author Contributions:** Methodology, Y.W. and Y.Z.; software, Y.Z.; validation, Y.W. and Y.Z.; formal analysis, Y.W.; investigation, Y.Z.; resources, L.W.; data curation, Y.W.; writing—original draft preparation, L.W.; writing—review and editing, Y.Z.; visualization, L.W.; project administration and research resource, R.L. All authors have read and agreed to the published version of the manuscript.

**Funding:** Funding was provided by the 2023 College Students Innovation and Entrepreneurship Training Program, project number S202310497099.

**Institutional Review Board Statement:** Not applicable.

**Informed Consent Statement:** Not applicable.

**Data Availability Statement:** Not applicable.

**Conflicts of Interest:** The authors declare no conflict of interest.

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
