# Peer review of "Design of a Four-Axis Robot Arm System Based on Machine Vision"

_applsci, doi:10.3390/app13158836_

Round 1

Reviewer 1 Report

  1. Abstract and conclusion needed analytical information. Abstract part required more technical information.
  2. Citation format is wrong
  3. Figure 1. System architecture of the robotic arm system is not clear. What is the need for this diagram?
  4. In page 10 line 242 author says “ The robotic arm used in this paper is based on the open source project 20sffactory, which has been improved”. What is 20sffactory?
  5. There is no technical details related to the design robot found in the manuscript. Author says stepper motor. But there is no data.
  6. Figure 10 The functional structure diagram of the system is meaningless. All the information are generic without technical data.
  7. The methodology part has been written in generic. Rewrite the methodology.
  8. In Upper computer implementation “Based on the above functional requirements, the Arduino ega2560+ramps1.4+A4988+stepper motor was chosen as the lower-level implementation of the robotic arm. In terms of the underlying code implementation, it mainly includes the communication program, the motion solver layer, and the motor motion control layer”. What is Arduino ega2560+ramps1.4+A4988+stepper from the above statement?

Need improvements

Author Response

Hello, we write the reply in the attachment, please check

Reviewer 2 Report

This study presents a vision guided based four-axis robotic arm embedded system, by introducing the target detection algorithm and deploying it to the embedded device, targeting towards the reduction of the robotic arm manufacturing cost and power consumption while meeting the control requirements.

Additionally, a comprehensive design of intelligent robotic arm control system is provided with the associated experimental results. The topic is interesting, is stimulating and relevant given the current context.

Starting from the introduction the authors are citing 3 studies in order to list the problems arising out of similar designs, it would be more supportive if they had included more citations mentioning the same problems. Additionally, at line 66 is written “To solve the above problems, a vision-guided intelligent robotic arm control” I would suggest that the authors write a small list numbering the problems detected, (1.5 lines for each problem) as the reader is lost and goes back in order to recall the detected specific problems.

I would suggest that the authors cite YOLOV7, Faster- 81RCNN and SSD even thought are well-known algorithms for further information of the readers who are not so familiar, (considering that these tools are basic tools in this study).

The methodology followed is clearly written, including the appropriate figures and the focus of this paper is specific. The objectives of the study are understandable.  The authors should discuss a little the potential and limitations (If ANY) of the study,

Author Response

(The authors gave the same response as above.)

Round 2

Reviewer 1 Report

All the points were addressed